# Epitranscriptomics of Ischemic Heart Disease—The IHD-EPITRAN Study Design and Objectives

**DOI:** 10.3390/ijms22126630

**Published:** 2021-06-21

**Authors:** Vilbert Sikorski, Pasi Karjalainen, Daria Blokhina, Kati Oksaharju, Jahangir Khan, Shintaro Katayama, Helena Rajala, Satu Suihko, Suvi Tuohinen, Kari Teittinen, Annu Nummi, Antti Nykänen, Arda Eskin, Christoffer Stark, Fausto Biancari, Jan Kiss, Jarmo Simpanen, Jussi Ropponen, Karl Lemström, Kimmo Savinainen, Maciej Lalowski, Markku Kaarne, Mikko Jormalainen, Outi Elomaa, Pertti Koivisto, Peter Raivio, Pia Bäckström, Sebastian Dahlbacka, Simo Syrjälä, Tiina Vainikka, Tommi Vähäsilta, Nurcan Tuncbag, Mati Karelson, Eero Mervaala, Tatu Juvonen, Mika Laine, Jari Laurikka, Antti Vento, Esko Kankuri

**Affiliations:** 1Department of Pharmacology, Faculty of Medicine, University of Helsinki, 00014 Helsinki, Finland; vilbert.sikorski@helsinki.fi (V.S.); daria.blokhina@helsinki.fi (D.B.); eero.mervaala@helsinki.fi (E.M.); 2Heart and Lung Center, Helsinki University Hospital, 00029 Helsinki, Finland; pasi.karjalainen@hus.fi (P.K.); kati.oksaharju@hus.fi (K.O.); helena.rajala@hus.fi (H.R.); satu.suihko@hus.fi (S.S.); suvi.tuohinen@hus.fi (S.T.); kari.teittinen@hus.fi (K.T.); annu.nummi@hus.fi (A.N.); antti.nykanen@hus.fi (A.N.); christoffer.stark@hus.fi (C.S.); fausto.biancari@hus.fi (F.B.); jan.kiss@hus.fi (J.K.); jarmo.simpanen@hus.fi (J.S.); jussi.ropponen@hus.fi (J.R.); karl.lemstrom@hus.fi (K.L.); markku.kaarne@hus.fi (M.K.); mikko.jormalainen@hus.fi (M.J.); peter.raivio@hus.fi (P.R.); sebastian.dahlbacka@hus.fi (S.D.); simo.syrjala@hus.fi (S.S.); tiina.vainikka@hus.fi (T.V.); tommi.vahasilta@hus.fi (T.V.); tatu.juvonen@hus.fi (T.J.); mika.laine@hus.fi (M.L.); antti.vento@hus.fi (A.V.); 3Tampere Heart Hospital, Tampere University Hospital, 33520 Tampere, Finland; jahangir.khan@sydansairaala.fi (J.K.); jari.laurikka@sydansairaala.fi (J.L.); 4Folkhälsan Research Center, 00250 Helsinki, Finland; shintaro.katayama@folkhalsan.fi (S.K.); outi.elomaa@helsinki.fi (O.E.); 5Graduate School of Informatics, Department of Health Informatics, Middle East Technical University, 06800 Ankara, Turkey; arda.eskin@metu.edu.tr; 6Heart Center, Turku University Hospital and Department of Surgery, University of Turku, 20521 Turku, Finland; 7Research Unit of Surgery, Anesthesiology and Critical Care, University of Oulu, 90014 Oulu, Finland; 8Clinical Biobank Tampere, Tampere University Hospital, 33520 Tampere, Finland; kimmo.savinainen@pshp.fi; 9Helsinki Institute of Life Science (HiLIFE), Meilahti Clinical Proteomics Core Facility, Department of Biochemistry and Developmental Biology, Faculty of Medicine, University of Helsinki, 00290 Helsinki, Finland; maciej.lalowski@helsinki.fi; 10Institute of Bioorganic Chemistry, Polish Academy of Sciences, Department of Biomedical Proteomics, 61-704 Poznan, Poland; 11Chemistry Unit, Finnish Food Authority, 00790 Helsinki, Finland; pertti.koivisto@ruokavirasto.fi; 12Helsinki Biobank, Hospital District of Helsinki and Uusimaa, 00029 Helsinki, Finland; pia.j.backstrom@hus.fi; 13Department of Chemical and Biological Engineering, College of Engineering, Koç University, 34450 Istanbul, Turkey; ntuncbag@metu.edu.tr; 14School of Medicine, Koç University, 34450 Istanbul, Turkey; 15Institute of Chemistry, University of Tartu, 50411 Tartu, Estonia; mati.karelson@ut.ee

**Keywords:** biomarkers, epitranscriptomics, ischemic heart disease, N^6^-methyladenosine, m^6^A, adenosine-to-inosine, A-to-I, RNA modifications

## Abstract

Epitranscriptomic modifications in RNA can dramatically alter the way our genetic code is deciphered. Cells utilize these modifications not only to maintain physiological processes, but also to respond to extracellular cues and various stressors. Most often, adenosine residues in RNA are targeted, and result in modifications including methylation and deamination. Such modified residues as *N*-6-methyl-adenosine (m^6^A) and inosine, respectively, have been associated with cardiovascular diseases, and contribute to disease pathologies. The Ischemic Heart Disease Epitranscriptomics and Biomarkers (IHD-EPITRAN) study aims to provide a more comprehensive understanding to their nature and role in cardiovascular pathology. The study hypothesis is that pathological features of IHD are mirrored in the blood epitranscriptome. The IHD-EPITRAN study focuses on m^6^A and A-to-I modifications of RNA. Patients are recruited from four cohorts: (I) patients with IHD and myocardial infarction undergoing urgent revascularization; (II) patients with stable IHD undergoing coronary artery bypass grafting; (III) controls without coronary obstructions undergoing valve replacement due to aortic stenosis and (IV) controls with healthy coronaries verified by computed tomography. The abundance and distribution of m^6^A and A-to-I modifications in blood RNA are charted by quantitative and qualitative methods. Selected other modified nucleosides as well as IHD candidate protein and metabolic biomarkers are measured for reference. The results of the IHD-EPITRAN study can be expected to enable identification of epitranscriptomic IHD biomarker candidates and potential drug targets.

## 1. Introduction

### 1.1. Ischemic Heart Disease

With 8.9 million yearly deaths worldwide, ischemic heart disease (IHD) is the leading cause of mortality [1]. It develops due to atherosclerosis, a process involving lipid and immune-cell buildup into coronary arteries forming calcified deposits, which eventually restrict blood flow to the myocardium [2]. These deposits have a tendency to rupture and occlude blood flow, thus causing myocardial infarctions (MIs). While timely revascularization can alleviate the damage to the myocardium relying on blood supply from the occluded artery, often the dead and injured tissue is replaced by a scar. With the heart devoid of any major regenerative ability, the processes of fibrosis and progressive stiffening often gain dominance and deplete the myocardium’s functional reserve [3]. Initially, the compensatory remodeling mechanisms, including cardiac myocyte hypertrophy, in other areas, can make up for the lost function. However, their extensive activation ultimately leads to structural and functional defects, collectively called maladaptive hypertrophy, compromising the heart’s filling and pumping ability. The resulting heart failure (HF) is a syndrome associated with high morbidity and mortality [4,5]. Disturbingly, IHD can exist asymptomatically without any signs of its existence, only to manifest as a sudden death [6]. Hence, the need for biomarkers capable of exposing IHD is apparent.

The quest for IHD biomarkers has yielded multiple candidates including systemic markers of inflammation and altered metabolism [7,8,9,10,11,12]. However, widespread use of these biomarkers is either limited or preliminary [13,14]. Currently, the diagnostic guidelines focus on the evaluation of symptoms, especially exertional chest pain (*Angina pectoris*) and dyspnea, determination of risk factors, such as age, sex, smoking, dyslipidemias, hypertension, diabetes, and family history of cardiovascular disease and risk factors-based calculators, such as Systematic Coronary Risk Estimation (SCORE) [15,16,17].

The current risk factors and validated risk calculators provide epidemiology-based estimations for the long-term emergence of “hard” IHD-related outcomes (e.g., fatal IHD, MI, stroke), which vary both in time and across cultures [18]. Direct assessment of a disease-associated or disease mechanism-coupled circulating biomarker could promote better-timed, targeted, and more personalized secondary prevention. Moreover, a biomarker capable of mirroring disease progression and regression would be invaluable for the evaluation of therapy efficacy. At best, it could even detect disease in asymptomatic individuals considered at risk of IHD with identical risk factor profiles.

### 1.2. Epitranscriptomics

On a molecular level, the bases of a newly transcribed RNA strand undergo extensive modifications both in the nucleus and cytoplasm. These epitranscriptomic RNA modifications have been identified as regulators of, for example, RNA splicing, silencing, localization, and stability that mediate or regulate a variety of processes involved in tissue homeostasis and disease [19]. In adenosine, the NH2-group at the purine ring’s sixth position may undergo enzymatic deamination to yield inosine (A-to-I) or methylation resulting in formation of N6-methyladenosine (m^6^A). Currently, these A-to-I and m^6^A modifications stand out as the most studied and abundant epitranscriptomic alterations, the extent and effects of which are governed by a variety of proteins (Figure 1) [20,21,22,23,24,25]. However, many other less charted and abundant such modifications have also been identified, including N^1^-methyladenosine (m^1^A), N^5^-methylcytosine (m^5^C), N^7^-methylguanosine (m^7^G), N^6^,2′-*O*-dimethyladenosine (m^6^A_m_), and pseudouridine (Ψ) [20].

Silencing or overexpression of enzymes controlling m^6^A abundance has revealed the role of m^6^A in driving immune reactivity, proliferation, apoptosis, and many intracellular processes including mRNA splicing, translation, and degradation [20,26], as well as miRNA biogenesis [27]. Moreover, reports from diverse fields of research [28,29,30,31], and in an array of cardiovascular pathologies [32,33,34,35,36,37,38,39,40,41,42,43,44,45,46,47,48], provide evidence of m^6^A as a master post-transcriptional regulator.

Since inosines pair with cytosines instead of uracils, the A-to-I modification is capable of diversifying the transcriptome [22]. Like m^6^A, A-to-I editing contributes to RNA stability and innate immunity, but also regulates RNA splicing as well as miRNA biogenesis and function [22,49,50]. Changes in the RNA A-to-I modification landscape have been associated with pathologies including cancer, neurological impairment [49], and cardiovascular disease [25,51,52,53,54,55,56].

### 1.3. Rationale and Goals

The hypothesis of the Ischemic Heart Disease Epitranscriptomics and Biomarkers (IHD-EPITRAN) study states that specific features of IHD are mirrored as epitranscriptomic modifications in the circulating blood RNAs. For example, these changes can manifest as alterations of m^6^A and A-to-I abundance or changes in their decoration patterns or loci within an RNA transcript (Figure 2). Such rationale is suggested by associations between the cellular and molecular processes governed by these modifications and the pathophysiology of IHD. Firstly, formation and dynamics of atherosclerotic plaques could contribute with epitranscriptomic signals into blood since they are the pathophysiological “hotspots” of IHD. They have characteristic features including active inflammatory cell proliferation, altered cellular and paracrine signaling microenvironments, and changes in the luminal presentation of blood cell response-modifying structures [57,58,59,60,61]. Secondly, the myocardium suffering from and responding to varying degrees of ischemia can either directly through e.g., shedding or production of RNA-packed extracellular vesicles (EVs), or priming and altering the patrolling blood cells’ responses contribute to epitranscriptomic cellular or cell-free RNA signature changes in blood. Thirdly, IHD promoting systemic responses, of which for example efflorescence of clonal hematopoiesis (CH) in bone marrow [62,63,64,65,66] and splenic hematopoiesis seeding proinflammatory monocytes [63,67,68] can be expected to alter blood epitranscriptomic signatures. Mechanistically, these postulations are suggested from notions that: (1) some leukocytes (1a) do exit the plaques [69], and (1b) oscillate between circulation and ischemic myocardium [70,71], (2) monocytosis has been independently associated with stable IHD and MI [72,73], (3) m^6^A has been shown to partake in dendritic cell (specialized monocytes) activation [74], (4) METTL3-mediated m^6^A-hypermethylation seems to act as a downstream elicitor of atherogenesis in vascular endothelium in response to disturbed flow and oscillatory shear stress [75], (5) the plaques, juxtaposed platelets, and ischemic myocardium are known to shed EVs to circulation encasing unique miRNAs [59,76,77,78,79,80,81], (6) miRNAs in such EVs have recently been shown to be epitranscriptomically modified [82], (7) epitranscriptomic and epigenetic regulators (7a) are often noted as CH driver mutations [83], and (7b) are pivotal for proliferation of hematopoietic stem cells (HSCs) [31,84,85,86,87,88]. Based on this rationale, the IHD-EPITRAN study aims to identify novel epitranscriptomic biomarkers and drug therapy targets for IHD from blood (Table 1).

## 2. Materials and Methods

### 2.1. Overview

The IHD-EPITRAN study recruits patients with both acute and chronic manifestations of IHD, non-IHD cardiac valve pathology, and healthy controls without IHD. The ethics review board at Helsinki University Hospital (HUS) approved the study protocol (Dnr. HUS/1211/2020). The study is registered at ClinicalTrials.gov (NCT04533282). Written informed consent is acquired from all patients before recruitment. Patient recruitment and follow-up began in November 2020. At the time of submission of this manuscript, the study had enrolled 40 patients. The results are expected to be published between 2022–2026. The study will be conducted following the Declaration of Helsinki on Ethical Principles for Medical Research Involving Human Subjects [89].

### 2.2. Study Groups, Recruitment and Exclusion Criteria

The IHD-EPITRAN study will recruit 50 patients in each of the four study groups to reach its first subgoal of a total of 200 patients. However, the study is expected to continue after the completion of this first phase to increase cohort sizes and study power over time (www.ihd-epitran.com, accessed on 24 May 2021).

The first study cohort (I) consists of patients admitted to the cardiac care unit (CCU) and recruited within 72 h of revascularization via percutaneous coronary intervention (PCI) for MI with ST-elevations (STEMI). The second study cohort (II) consists of patients with chronic coronary syndrome (CCS, typically of category 1: suspected chronic IHD due to anginal symptoms [12]). These are patients with persistent IHD symptoms scheduled to undergo elective coronary artery bypass grafting (CABG) operations. The third study cohort (III) consists of non-IHD patients undergoing aortic valve replacement surgery (AVR) for aortic valve stenosis (AVS). The fourth study cohort (IV) consists of patients diagnosed as negative for IHD with coronary artery computerized tomography angiography (CCTA) imaging. The former two cohorts (I-II) represent patients with IHD while the latter two cohorts (III and IV) represent non-IHD control patients. A chronologic and cross-sectional summary of the study cohorts is provided in Figure 3.

While patients in cohort I will be recruited during the hospitalization period, the cohorts II and III will be recruited during the preoperative visit. Patients in cohort IV will be either (1) preselected based on received referrals by the outpatient-treating physicians to CCTA imaging or (2) contacted after CCTA imaging if it has been negative for IHD within the last 3 months and exclusion criteria are met (Table 2).

All recruited patients will be invited for an echocardiography visit (Section 2.3.4). The study information sheets will be sent well beforehand to ensure familiarization of the study for the candidate participants when possible (cohorts II and IV) and delivered as soon as practical according to the clinical situation in CCU for patients in cohort I. Patients aged 18–80 years and meeting the cohort-specific descriptions are eligible for participation in the study. The exclusion criteria and their rationales are presented in detail in Table 2.

### 2.3. Study Measures

The laboratory methods described below (Section 2.3.5, Section 2.3.6, Section 2.3.7, Section 2.3.8, Section 2.3.9 and Section 2.3.10) for the study sample analytics shape a principal protocol pipeline for the study. However, the authors reserve the rights to alter utilized analysis methodologies to enable both smooth and most appropriate adaptations to the evolving field of epitranscriptomics regarding its methodology as well as pathophysiologial insight.

#### 2.3.1. Baseline Morbidity

To manage the confounding variables, a general baseline morbidity assessment will be performed. The assessment will be carried out using a complementary approach that includes a patient information system search for comorbidities (e.g., hypertension, diabetes, kidney failure, inflammatory conditions, and chronic obstructive pulmonary disease), structured patient interview (Section 2.3.2), and a case report form (CRF) fill-out that includes traditional Framingham cardiovascular risk factors [90], supplemented with body mass index (BMI), personal history of MI, or family history of IHD, determination of SCORE 10-year risk estimates for fatal IHD (for the non-IHD cohorts III-IV) [17], medication list review, coronary imaging (Section 2.3.3), myocardial imaging (Section 2.3.4), and follow- up (Section 2.3.5). Data on patient medication will be curated and analyzed using the InnoLIMS^®^ Medical (Innovatics Ltd., Helsinki, Finland) or equivalent software based on anatomical therapeutic chemical (ATC) classification using the defined daily dose (DDD) values enabling dosage comparisons.

#### 2.3.2. NYHA, CCS and SF-36

*Angina pectoris* and exertional dyspnea, the two central symptoms of IHD, will be graded pre- and postoperatively at the three-month follow-up using the standardized classifications systems from the Canadian Cardiovascular Society (CCS) and New York Heart Association (NYHA) [91]. Further, the 36-Item Short Form Health Survey (SF-36) is used to assess subjective participant morbidity [92].

#### 2.3.3. Coronary Angiography, Computed Tomography Angiogram, SYNTAX Score

An invasive coronary angiography (ICA) will be performed for all participants in study cohorts I–II and for most in AVR control cohort III based on clinical details (mainly for those >50 years of age). ICA will be used to identify the precise site and quality of coronary thrombosis and to focus the PCI revascularization on the cohort I. On the other hand, for the cohort II, ICA will be performed to identify the main culprit segment(s) and to guide the upcoming CABG operation. Critically, based on a clinical risk stratification, either ICA or CCTA will be performed as a screening test to expose either obstructive or non-obstructive IHD, respectively, prior the AVR operation in the AVS control cohort III. All patients in the control cohort IV will undergo CCTA imaging.

While a zero Agatston coronary artery calcium score [12] from CCTA imaging will be required for inclusion to the IHD-EPITRAN study (all in cohort IV, some in III), a synergy between percutaneous coronary intervention (PCI) with Taxus and coronary artery bypass surgery (CABG) (SYNTAX) will be calculated from all ICA results, providing symptom-independent estimates of the complexity of IHD [93]. Moreover, the culprit segment, number, and types of stents or bypasses during PCI or CABG in cohorts I and II, respectively, will be recorded.

#### 2.3.4. Echocardiography, 3-Month Control Visit, and Follow-Up Period

As the development of more sophisticated echocardiographic techniques has revolutionized the field of cardiology and cardiac surgery by offering relevant dynamic insights to both cardiac anatomy and function, IHD-EPITRAN will assess all participants in all cohorts with transthoracic echocardiography (TTE). Critically, the non-IHD control patients in cohort IV will also undergo TTE recording. In this way, valuable functional reference values are obtained.

The TTE recordings will be performed with prespecified acquisition methods by designated cardiologists [94]. The TTE recordings will include both anatomical and functional assessments of atria, valves, and ventricles as well as recording the presence or absence of pericardial effusion, thrombus, and aneurysm. Moreover, the echocardiographic raw data will be exported and stored for further analyses, such as strain and strain rate measurements.

In addition, the 3-month follow-up visit includes a morbidity evaluation with the use of CCS and NYHA and SF-36 questionnaires (Section 2.3.2) alongside routine clinical anamnesis and status. Moreover, the CABG cohort II will undergo ankle-brachial indexing (ABI) to record any asymptomatic peripheral artery disease (PAD).

A six-month prospective follow-up period starting from recruitment will be incorporated. Additionally, a six-month retrospective evaluation will be performed preceding the moment of recruitment. Hospital admission rates with primary cause(s), medication changes, determination of Major Adverse Cardiac and Cerebrovascular Events (MACCE), and all-cause mortality for follow-up make up the parameters assessed both during retrospective evaluation and prospective follow-up.

Hospital admissions regarded as cardiovascular-based will be recorded using the International Statistical Classification of Diseases and Related Health Problems (ICD-10). The IHD-EPITRAN study will use a modified MACCE definition adopted from the ITALIC trial [95] with a composite primary outcome including cardiovascular death, MI, acute revascularization, stroke, transient ischemic attack (TIA), major bleeding (The Bleeding Academic Research Consortium [BARC] classes 3–5), or other hospitalization due to ischemic cardiovascular cause [96].

All-cause mortality with the ICD-10 codes for both underlying and immediate causes of death from the death certificates from the six-month follow-up will be documented with a focus on deaths considered as cardiovascular-related. Cardiovascular medication changes with ATC codes and DDDs will be recorded and analyzed with InnoLIMS^®^ Medical or equivalent software (Section 2.3.1). The medication data yielded will subsequently be used to model either an improved (reduced medication or dosage) or worsened (added medication or increased dosage) disease state.

#### 2.3.5. Study Blood Samples

While a set of collected blood samples will be stored directly by the study personnel, another set is stored to respective collaborative biobanks of each participating clinical center with partial sample-derived aliquot reservation. The non-biobank study blood samples will include TEMPUS^TM^ whole blood samples (3 mL × 5) and EDTA blood-derived (9 mL × 3) plasma aliquots. The biobank samples will include EDTA blood-derived (10 mL × 1) plasma and buffy coat aliquots as well as blood-derived (10 mL × 1) serum aliquots. In addition to the above-described set of biobank samples, if needed, a single 3 mL EDTA blood sample is collected for DNA extraction. The samples will be collected twice in study cohorts I–III and once in cohort IV (Figure 3).

The first blood samples for cohort I (acute STEMI phase) will be collected within 72 h after PCI, but the goal is within the first 24 h. While the IHD-EPITRAN blood samples will be prioritized for processing within 1 to 1.5 h, the processing time span for the biobank samples can be expected to extend up to 8 h due to practical restraints. A summary of the study samples, their general processing and principal analytical methods with corresponding goals pursued are provided in Figure 4.

Whole blood samples—Cellular RNA will be derived from the TEMPUS^TM^ blood sample tubes designed to preserve whole blood RNA against the otherwise rapid extracorporeal degradation. First, the acquired TEMPUS^TM^ blood samples will be vigorously shaken for 10 s and stored at <−70 °C until further processing. Second, the whole-blood total RNA will be isolated from the rest of the sample material according to the manufacturer’s instructions (e.g., TEMPUS^TM^ Spin RNA Isolation Kit, ThermoFisher, Waltham, MA, USA) before storage for further processing (Section 2.3.7).

Plasma samples—Cell-free RNA (cfRNA) and surrogate cardiovascular biomarkers will be assessed from the IHD-EPITRAN sample pool’s EDTA blood-derived plasma aliquots. For cfRNA analysis, we have performed a confirmatory pilot investigation following a previously described protocol by Shiotsu et al. [97]. Briefly, plasma will be separated and aliquoted (900 μL × 10) into RNase-free tubes preloaded with RNA-stabilizing MLP lysis buffer (270 μL each; catalog ref. 740,365.75, Macherey-Nagel, Düren, Germany) and stored at <−70 °C. The cfRNA will be extracted from MLP plasma utilizing a purification kit according to the manufacturer’s instructions (e.g., Nucleospin^®^ miRNA Plasma, Macherey-Nagel, Germany). Based on our initial testing of this protocol, each preloaded aliquot (900 μL + 270 μL) yielded approximately 17 ng of purified cfRNA, as measured with Qubit^®^ microRNA Assay kit (ThermoFisher Scientific, USA). The cfRNA will then be quantitatively (Section 2.3.8) and qualitatively (Section 2.3.9) assessed for m^6^A modifications and A-to-I editing events.

The plasma used for the measurement of surrogate cardiovascular biomarkers will be aliquoted (500 μL × 4–10) without MLP buffer into RNase-free tubes and stored at <−70 °C. The biobank EDTA-blood sample (10mL) will be processed and separated into both buffy coat (Helsinki, 500 μL × 1 and 300 μL × 1; Tampere, single aliquot) and plasma aliquots (450 μL × 4) and stored at <−70 °C. The serum blood sample will be phase-separated into 8 aliquots (400 μL) in total. Three aliquots of both serum and plasma are reserved for the study. While either the buffy coat or whole EDTA blood (3 mL) will be used for DNA extraction at Helsinki and Tampere biobanks, respectively (Section 2.3.9), the plasma and serum aliquots will primarily be used for assessment of metabolites and targeted proteomics (Section 2.3.10). The rest of the buffy coat (after DNA extraction) will be stored (<−70 °C) for later use, such as leukocyte-specific RNA extraction.

Metabolite surrogate cardiovascular biomarkers—Metabolites will be principally assessed from biobank plasma aliquots due to their longer processing timespan suboptimal for RNA analytics. Analysis of metabolites may be extended to untargeted metabolomics or focus on targeted selected metabolites (Figure 4). For example, based on a metabolomic profiling of human plasma, TMAO, a gut microbiota-derived metabolite from choline, has recently been suggested to stimulate atherosclerosis and platelet hyperreactivity [11]. Similar to TMAO, another gut microbiota-derived plasma metabolite, phenylacetylglutamine (PAG), has recently been identified to associate with increased composite risk of MACE (MI, Stroke, Death) [12].

Non-metabolite surrogate cardiovascular biomarkers—The IHD-EPITRAN will correlate acquired IHD-related blood epitranscriptomes with surrogate biomarkers for cardiovascular diseases, such as hsCRP [7], sST2 [8], copeptin [10], blood cholesterols and N-terminal pro-hormone B-type natriuretic peptide (NT-proBNP) [98]. These will be analyzed from non-MLP plasma with specific commercial, principally ELISA-based, laboratory kits. An online-only data Appendix A lists a selected summary of articles assessing the biomarkers measured in the IHD-EPITRAN [10,11,12,13,99,100,101,102,103,104,105,106,107,108,109,110,111,112,113,114].

#### 2.3.6. Right Atrial Appendage Tissue Samples

The right atrial appendage (RAA) is often manipulated during cardiac surgery to assemble bypass circulation. Moreover, RAA has been evaluated as a safe source for cardiac tissue for both diagnostic and therapeutic applications [115]. During the IHD-EPITRAN, a small RAA tissue sample, the size of which will be guided by clinical characteristics, will be collected from the study participants in cohorts II and III to enable an organ-specific characterization of the RNA epitranscriptomes (Figure 4). The perioperatively collected RAA tissue sample will be divided into three pieces that will be stored in (1) RNAlater solution (e.g., AM7021, ThermoFisher Scientific Inc., Waltham, MA, USA) with an overnight incubation (4 °C) prior to storage (<−70 °C), (2) formalin (4%) for two weeks and then in ethanol (70%; Section 2.3.9) to prevent overfixation or (3) snap-frozen preferably with isopentane immersion and liquid nitrogen and stored (<−70 °C). The RNA isolation of the RNA later-stored RAA samples is done as previously [115]. A brief consideration of the RNA fractionation pipeline is provided in Section 2.3.7. The piece of RAA tissue snap-frozen in isopentane immersion will be a well-suited material for many histologic and omics applications, such as untargeted proteomics and spatial RNA transcriptomics (Section 2.3.9 and Section 2.3.10, respectively).

#### 2.3.7. RNA-Stabilized Blood and RAA Tissue RNA Fractionation

The IHD-EPITRAN study will characterize the various RNA-species-specific epitranscriptomic alterations in blood in IHD. Prior to the RNA fractionation itself, sample DNA is either degraded with DNAse treatment according to the kit manufacturers’ instructions (plasma cfRNA and RAA tissue RNA extraction) or phase-separated via RNA pelleting (RNA-stabilized TEMPUS^TM^ blood extraction). Further, extracted total RNA will be subjected to fractionation to achieve distinct sequencing datasets for protein-coding messenger RNAs (mRNAs), small RNAs with regulatory properties (e.g., micro-RNAs and long non-coding RNAs), as well as ribosomal and transfer RNAs. Due to both pragmatic reasons and probable future developments, the specific RNA fractionation protocol is left undecided, but will include most probably the following: depletion or blockage of globin RNAs, depletion of ribosomal RNAs, as well as a poly-A capture of mRNAs. Moreover, since the third-generation direct long-read nanopore sequencing (Section 2.3.9) requires either a poly-A or custom tail for the recognition by a sequencing adapter, poly-A tailing or custom tailing of fractionated RNAs will be performed prior to sequencing (Section 2.3.9).

#### 2.3.8. Quantitative RNA Modification Analysis with UHPLC-LC-MS/MS

A quantitative analysis, with an emphasis on both the m^6^A and inosine (I), for the eight epitranscriptomic modifications (e.g., m^1^A, m^6^A, m^6^A_m_, ac^6^A, m^5^C, m^7^G, pseudouridine [Ψ] and I) will be performed utilizing an ultra-high-performance triple quadrupole liquid chromatography tandem mass spectrometry (UHPLC-LC-MS/MS) method as previously described [116].

#### 2.3.9. Qualitative Analyses of RNA Modifications

Sequencing—To expedite the various sequencing tasks in the IHD-EPITRAN study, the sequencing services will be purchased from third party sequencers after competitive tendering. The raw data will be acquired and analyzed by the IHD-EPITRAN Consortium (Section 2.5). While methylated immunoprecipitation (meRIP) [117,118] and direct long-read sequencing [119] are the two most widely used examples of the RNA sequencing protocols utilized for m^6^A identification, others have also been described [120,121].

The final sequencing protocol will be defined during the course of the study. Several techniques, including untargeted epitranscriptomic sequencing methods, to measure m^6^A are available. Their strengths and weaknesses are listed in Appendix A [20,116,117,118,119,120,121,122,123,124,125,126,127,128,129,130,131,132,133,134,135,136,137,138,139,140,141,142]. First, the IHD-EPITRAN study plans to utilize, based on competitive tendering, one of the various mainstay second generation sequencing platforms for detecting the epitranscriptomically modified transcripts for initial insight [117,118,120,121]. Of these, albeit shown to come with its technical limitations especially regarding in-sample replicate reproducibility [122], meRIP-sequencing is preferred as the most widespread m^6^A sequencing methodology used to date [117,118]. Such sequencing can be performed for all study samples of the IHD-EPITRAN (RNA-stabilized TEMPUS^TM^ whole blood and its RNA fractions, EDTA-blood derived plasma, and RAA tissue) to acquire a fraction-wise data of m^6^A-enriched RNA transcripts to be compared against the novel third generation, or direct long-read, sequencing with its providable single transcript-level in situ m^6^A datamaps [119]. In more detail, nanopore sequencing (Oxford Nanopore Technologies, Oxford, UK) is the most well-established prototype of such direct sequencing basing its function in carefully designed protein nanopores embedded within a semipermeable membrane compartmentalizing loaded and tailed native sample RNAs initially to the sample chamber medium [119]. Next, electrical potential difference (voltage) is imposed across this nanoporous membrane with negative charge on the side of the sample medium. These sophisticated nanopores not only act as the sole passages of the ionic currents through the membrane, and thus create a recordable reference signal, but also allow the passthrough of the native sample RNAs crucially preserved in terms of their contained base modifications. As native RNAs pass through the nanopores, characteristic disruptions from each consecutive RNA base, both unmodified and modified, are formed and recorded relative to the reference signal [119]. These disruptions are ultimately decodable in silico with a base calling algorithm EpiNano (https://github.com/enovoa/EpiNano, accessed on 18 June 2021) that is currently validated for m^6^A with ~90% accuracy, but expectable to become more precise as well as putative to expand to other modifications as well (e.g., m^5^C, and m^7^G). In addition, either exome or genome sequencing will be performed from buffy coat or EDTA blood extracted DNA (Section 2.3.5) for comparison with respective RNA sequence to pinpoint A-to-I editing events [143].

Histological staining—Considering the encouraging evidence suggesting detection of m^6^A in the near future with fluorescence in situ hybridization (FISH) [144], we aim to perform a m^6^A-targeted staining to reveal the in situ localization of modified RNAs in histological sections of the RAA tissue as a supplementary insight to the UHPLC-LC-MS/MS and sequencing. This approach could enable us to pinpoint the varied m^6^A expression associated with IHD to the distinct cell groups present in the human heart (i.e., vasculature, cardiac interstitium, and cardiomyocytes). Furthermore, since the formalin/ethanol-stored and isopentane snap-frozen RAA tissue pieces constitute high-quality material with regard to histological approaches, targeted protein immunohistochemistry, RNA in situ hybridization (e.g., RNAscope^®^, Advanced Cell Diagnostics, Inc., Bio-Techne Corporation, Minneapolis, MN, USA), and spatial RNA transcriptomics (e.g., Visium, 10× Genomics, Pleasanton, CA, USA), will be assessed for utilization to maximize the still sparse histological insight regarding epitranscriptomics in human heart in IHD. This can be achieved, for example, by staining for both the enzymes itself governing A-to-I editing and m^6^A and their respective mRNAs. Moreover, spatial transcriptome sequence comparison to the DNA sequence of an individual directly provides unprecedented histological A-to-I landscapes from human cardiac tissue.

#### 2.3.10. Proteomics

To augment IHD-biomarker discovery and provide referencing measures for the forthcoming epitranscriptomic alterations, the IHD-EPITRAN study will utilize proteomics applications for both biobank-stored plasma aliquots and snap-frozen RAA tissue. For plasma, the suitability of a targeted proteomics approach with a set of several hundreds of proteins, via application of a cutting-edge multiple reaction monitoring (MRM) and labeled peptide counterparts, is carefully assessed [145]. For the snap-frozen RAA tissues, an untargeted label-free mass spectrometry approach is primarily preferred [115,146].

### 2.4. Power Calculations, Cohort Comparisons, Outcomes, and Data Management

The group sizes were determined with a RNAseqPS tool designed for evaluating statistical power for sequencing studies [147]. Parameter values (false discovery rate [FDR] <0.05, total number of genes for testing 20,000, predicted prognostic genes 1500 with a minimum fold change threshold of 2 for differential expression and average read counts of 10 for prognostic genes) were derived from the applicable study regarding m^6^A during hypertrophy and HF [148]. A mean dispersion value of 0.215 was doubled (0.43) and applied for the calculations. The utilized mean was based on four assessed sequencing datasets with unrelated samples and dispersions ranging from 0.15 to 0.28 [149]. Based on the power analysis, *n* = 44 per cohort generated a power of 0.95. However, the groups sizes were fixed to *n* = 50 per cohort to address possible preanalytical errors at this initial phase. Analyses with smaller sample sizes will be first performed, followed by the larger sized analyses based on the first results and methodological optimization.

Currently, there are no reports addressing either gender or age dependent changes in neither blood nor muscle tissue epitranscriptomes. As such, the produced data in the IHD-EPITRAN study will be referenced and assessed against the typical common confounding variables, such as gender, age, comorbidities, and medication. If needed, the effects will be statistically adjusted. The IHD-EPITRAN consortium (Section 2.5) includes experts with statistics expertise and knowhow.

The study cohort comparisons used to acquire the study outcomes of the IHD-EPITRAN study are presented in Figure 5 and Table 3, respectively. The project Data Management Plan (DMP) is maintained on the webpage of the IHD-EPITRAN study (www.ihd-epitran.com).

### 2.5. Collaborators, the IHD-EPITRAN Consortium

The IHD-EPITRAN study has been designed in collaboration with project members consisting of cardiology and cardiac surgery specialist clinicians and scientists focusing on genomics, proteomics, RNA m^6^A, and A-to-I analytical methodologies and bioinformatics, and in silico, in vitro, and in vivo drug development. This interdisciplinary team, the IHD-EPITRAN Consortium, will enable the study to synergistically perform patient recruitment, sample collection, and preparation followed by detailed bioinformatic analyses of the epitranscriptomic m^6^A and A-to-I editing landscapes in circulating blood and RAA tissue during acute and stable IHD, AVS, and cardiac health. While HUS and the Department of Pharmacology from University of Helsinki (UH) share responsibility as the data controllers, the current collaborator organizations are listed in Table 4. The IHD-EPITRAN Consortium warmly welcomes interested clinical and scientific centers to participate in the study (www.ihd-epitran.com; see below).

### 2.6. Long-Term Follow-Up, IHD-EPITRAN Extensions

Long-term follow-up of the study cohorts for up to 10 years or more after recruitment can provide relevant insight additive to the current cohort comparisons and outcomes, which are presented in Figure 5 and Table 4, respectively. Specifically, regularly repeated blood sampling coupled with a suitably synchronized coronary and myocardial evaluation, recording of IHD exacerbation and new-onset IHD case data in cohorts I and IV, respectively, produces highly valuable samples suitable for epitranscriptomic scrutinization. Hence, analysis of such samples equipped with powerful clinical metadata hold potential to decipher epitranscriptomic candidate biomarkers for prognosticating long-term MACCE and asymptomatically developing, but potentially fatal, IHD. As such, we plan to continue the IHD-EPITRAN study after the end of active phase by calling the herein recruited participants for long-term follow-ups. This will be executed as distinct and later-named follow-up extension studies, named, for example, as IHD-3/5-EPITRAN and IHD-10-EPITRAN.

## 3. Expected Results

The IHD-EPITRAN study provides both quantitative and up-to single nucleotide in situ qualitative m^6^A and A-to-I focused epitranscriptomic datasets from whole blood, plasma, and RAA tissues (Section 2.3.8 and Section 2.3.9) from both meticulously characterized and prospectively recruited clinical cohorts representing IHD, AVS, and cardiac health (Section 2.2). Moreover, both MRM and untargeted proteomic profiles from plasma and snap-frozen RAA tissue, respectively, are acquired (Section 2.3.10) alongside comprehensive panels of plasma metabolite and non-metabolite surrogate CVD biomarkers (Section 2.3.5). Additionally, in situ histological landscapes of A-to-I editome and m^6^A governing enzyme expressions from clinical RAA tissue samples affected by both IHD pathophysiology and AVS pressure overload are produced (Section 2.3.9).

The controlled prospective cohort design of the study (Section 2.2, Figure 3) enables multiple clinically relevant aspects to be evaluated for the aforesaid datasets. For example, IHD vs. non-IHD distinctions, comparison of acute and chronic manifestations of IHD, therapy evaluation and long-term detection of IHD exacerbations, as well as newly arising IHD (Section 2.6) are both achievable and assessable regarding these datasets when referenced to the comprehensively recorded clinical metadata (Section 2.3.1, Section 2.3.2, Section 2.3.3 and Section 2.3.4). The arising key outcomes are listed in Table 3 and illustrated in Figure 5. Importantly, the study is expected to identify several IHD-specific candidate modified RNAs as well as possible IHD-associated (consensus) sequences that harbor epitranscriptomic alterations, as recently suggested from human failing hypertrophic myocardium [36]. Because RNA modifications other than m^6^A and A-to-I are to be quantitatively evaluated as well, the study results can help us to better understand also the other epitranscriptomic pathway contributions to IHD pathophysiology.

Furthermore, considering the rapid development of epitranscriptomic assessment methodologies (especially machine learning based modification-oriented base calling algorithms such as EpiNano), it is expectable that the current panel of qualitatively assessable RNA modifications from the third generation sequencing datasets, yet restricted to m^6^A, expands [119]. Hence, the sequencing datasets produced in the course of the IHD-EPITRAN could later be subjected to, for example, m^5^C (N^5^-methylcytosine)-targeted analysis. The study produces carefully recorded echocardiographic raw data, which can be used not only as a state-of-the-art reference guiding interpretation of the aforesaid epitranscriptome-oriented exploratory datasets, but also as a standalone source of insight when compared against other recorded clinical variables as well as CCTA and ICA imaging results. Lastly, the SYNTAX-scores for IHD complexity may be utilized to correlate intracohort epitranscriptomic profiles with the locations and extent, and thus indirectly activity, of coronary atherosclerosis.

## 4. Discussion

Despite advances in medical care, IHD remains the global leading cause of death [1]. Combined with the lack of validated blood biomarkers for exposing either early-stage or established IHD, the need to find IHD-specific biomarkers is evident [12]. More specifically, the current risk factors and calculators provide rigorously validated epidemiology-based risk estimates for the long-term (usually 10-year timespan) emergence of “hard” IHD outcomes (e.g., fatal IHD, MI, stroke), which, however, are subjects for considerable variation both in time and across cultures, as recently reviewed [18]. As such, they cannot directly expose, and thus differentiate, individuals harboring an asymptomatically progressing IHD from those devoid of such disease activity. Thus, the current estimation tools inevitably overestimate the future risk for IHD events for some while critically underestimating it for the others. This manifests as suboptimally targeted resources for patient counseling, self-care motivation, and medical interventions, i.e., secondary prevention. Hence, identification of blood biomarker(s) capable to mirror the IHD pathogenesis per se could better (1) stand time and changing lifestyles and (2) shift the diagnostic and therapeutic decision-making guidelines of asymptomatic patients from the current core dependence on epidemiology-based risk estimations towards increasingly individualized considerations. In practice, such shift could lead to both better-timed and targeted secondary prevention ultimately reducing the progression of IHD towards incurable, disabling and fatal late-stage complications, such as congestive HF. Furthermore, as originally proposed already by Wilson and Jungner in 1968, IHD fulfils most of the key criteria for a condition amenable to mass screening, including high disease prevalence, morbidity, and mortality, existing treatments, and secular development through precursor states (endothelial stress—leukocyte infiltration—atherogenic strands—obstructive plaques and ischemia—plaque instability—plaque rupture and infarction—tissue destruction—congestive HF) [150].

Although the study of RNA modifications, epitranscriptomics remains in its infancy, methodological breakthroughs of the last decade have enabled identification of these modifications with such accuracy that their large-scale screening is rational [117,118,119,120,121,143]. Encouragingly, research findings suggest both m^6^A and A-to-I to act as contributors or even potential initiators and drivers for several cardiovascular physiological and pathological processes including cardiogenesis, angiogenesis, hypertension, hypertrophy, atherosclerosis, ischemia, ischemia-reperfusion, fibrosis, HF, congenital heart disease, stroke, aneurysms, as well as cardiac repair and regeneration [25,32,33,34,35,36,37,38,39,40,41,42,43,44,45,46,47,48,51,52,53,54,55,56]. Remarkably, the first indication for coronary atherosclerosis to be reflected in the m^6^A content of mRNAs and long non-coding RNAs of peripheral mononuclear cells with suggested involvement in its pathophysiology has just recently been reported [151]. However, to the best of our knowledge, the IHD-EPITRAN is yet the first controlled prospective observational clinical study recording meticulous metadata to specifically support its goal to broadly address blood epitranscriptomics in IHD with diverse sample collection designed for the (near-) instant halting of RNA degradation from the moment of sample collection. As such, the IHD-EPITRAN study responds to the recently voiced “epitranscriptomic challenge” [152].

Etiologically, IHD develops due to atherosclerosis in coronary arteries with origins often tracing back to early adolescence [153]. As a currently understood initiating step, buildup of cholesterol and lipoproteins on sites of both disturbed blood flow and low shear stress induces—dependent on endothelial stress, local dendritic cells, chemokines, endothelium-expressed homing receptors, and platelets—homing and extravasation of various populations of leukocytes and their pro-inflammatory polarized enrichment at the site of such lesions [57,154,155,156]. On these sites, monocytes, hardwired to rapidly differentiate into inflammatory macrophages, ultimately transform into foam cells following their phagocytosis of subintimal lipids as a clearing attempt of such wrong mater at the wrong place [57]. However, macrophages simultaneously also secrete paracrine factors, such as netrin-1 [157], that hamper their effective egress from the lesions. As such, the plaque-residing macrophages keep proliferating and eventually die leading to on-site necrosis, formation of necrotic lipid cores with sustained inflammation, and production of both cytokines and reactive oxygen species (ROS) [58]. Such a pernicious milieu promotes an inflammatory response also in the plaque-lining endothelium. Increased endothelial permeability and homing receptor expression intensify leukocyte extravasation to the growing and weakening plaques giving rise to IHD and eventually MI [60]. Simultaneously, the paracrine signaling between myocardium and the newly dysfunctional coronary endothelium fall rapidly into disarray from its baseline reciprocal state. This is manifested not only as inappropriately increased production of vasoconstrictors, such as endothelin 1, but also by dampened production of critical vasodilators, such as nitric oxide, prostacyclin, and neuregulins promoting the deepening of myocardial ischemia [61,158]. The inflamed plaque-myocardial-microenvironment also disseminates systemic signals in the form of autonomous nervous system activation, circulatory cells, and soluble signaling molecules, such as cytokines, which induce atherosclerosis-accelerating responses in both immune and hematopoietic systems. Spleen and bone marrow, the two most studied extracardiac organs responding to these cues, overproduce polarized monocytes promoting further plaque invasion and increase the proliferation of HSCs accelerating the efflorescence proatherosclerotic clonal hematopoiesis, respectively [57,62,63,64,65,66,67].

The hypothesis of the IHD-EPITRAN study states that m^6^A and A-to-I modifications in blood-derived RNAs mirror atherosclerosis, IHD and MI. At cellular and molecular levels, these modifications have been suggested to have varying degrees of dynamics relative to the RNA transcript in which they reside. [159,160,161,162,163,164]. Moreover, several sites in a single transcript can be targeted for modification. Together, these provide molecular foundations for biomarker discovery, as these modification profiles could store sufficiently stable information.

To provide insight into the possible mechanistic linkages between m^6^A and A-to-I modification systems and IHD, we provide here a more detailed view into three putative mechanisms (Figure 2). First, ischemic myocardium may produce epitranscriptomically detectable signals to the bloodstream in the form of either ischemia-primed monocytes patrolling between myocardium and circulation (as suggested during cardiac homeostasis and MI [70,71]) or as secreted EVs. Indeed, ischemic myocardium is known to abundantly secrete EVs encasing cardiac-specific miRNAs with promising biomarker properties [76,77], and such EVs may also hold modified RNAs. This is suggested since m^6^A [27] and A-to-I [22,51,52] have been shown to exist in miRNAs and regulate their biogenesis and targets. Besides, a recent report provides a proof-of-concept for epitranscriptomic signal discovery from EVs, as the m^6^A deposition onto miR-19 during its biogenesis enhances its loading into EVs [82]. Second, similar EVs containing m^6^A or A-to-I decorated RNAs can also arise from biological sources relevant for IHD other than ischemic myocardium. Such assertion is conceptualized through knowledge that apoptotic leukocytes from coronary plaques, plaque-lining endothelium [59], platelets adhering such endothelium, and hypertension-induced circulating platelets shed EVs as a part of their paracrine signaling [78,79,80,81]. Furthermore, a recent report indicates the METTL3-mediated m^6^A-hypermethylation to act as a driver of the initiating atherogenic events in vascular endothelium subjected to both disturbed flow and oscillatory shear stress [75].

Thirdly, we postulate that the earlier suggested [63,64,65,66] and recently, in terms of causality, shaped [62] vicious cycle where atherosclerosis stimulates clonal hematopoiesis that circles back to further promote atherosclerosis through yet unknown, possibly cytokine-based, factors could seed leukocytes into bloodstream with characteristic phenotypes and m^6^A and A-to-I signatures mirroring the cycle. Further, we propose such signatures to be detectable by deep third generation sequencing of whole blood RNA extracts. These relatively weighty postulations suggesting detectable involvement of epitranscriptomics with the atherosclerosis-stimulated clonal hematopoiesis rest on several notions: (1) driver mutations for clonal hematopoiesis in HSCs, when present, often reside in epigenetic or epitranscriptomic targets [83], and (2) multiple governing enzymes of A-to-I [84] and m^6^A [31,85,86,87,88] have been shown to be essential for maintaining normal hematopoiesis. Finally, as proof-of-principles, emerging evidence suggests blood epitranscriptomes to act as potential source of biomarkers for coronary atherosclerosis and thus IHD [151], breast [165], gastric [166], and lung [167] cancers as well as for few other inflammation-centered pathologies than IHD, such as systemic lupus erythematosus [168] and rheumatoid arthritis [169].

Considering the multileveled entanglement of epitranscriptomics with IHD pathophysiology, that is even suggested as a driver of some crucial steps of its pathogenesis [75], these modifications could also provide a novel source of therapeutic drug targets for IHD. Indeed, small molecule ligands for METTL3 writer complex [116], FTO [170], and ALKBH5 [171] erasers have already been described by the members of the IHD-EPITRAN Consortium. Remarkably, just recently, a potent METTL3 inhibitor has also been reported with leukemia-repressing effects in vivo in mice, providing simultaneously an enchanting proof-of-principle and a seminal endeavor to thrust the door ajar into the yet uncharted realm of epitranscriptomics-based pharmacology in vivo—ultimately shifting eyes also increasingly towards the clinic [172]. Hence, both the existing and future breakthroughs regarding epitranscriptomic pharmacology is closely monitored and implemented accordingly to the mid/late phases of the IHD-EPITRAN study as a further applicatory dimension.

A list of the study strengths and limitations is given in Table 5. Major strengths of the IHD-EPITRAN study include: (1) wide interdisciplinary collaboration network (Table 3) that enables simultaneous participant recruitment to (2) the relevant IHD and non-IHD cohorts, (3) meticulous examination of the degree of coronary atherosclerosis, and (4) comprehensive recording of major confounding variables. As limitations, we cannot fully exclude the effects of systemic atherosclerosis and considering the unmet need to detect asymptomatic IHD, the current IHD-EPITRAN study protocol with its brief follow-up cannot identify biomarkers for subclinical IHD due to required morbidity. However, the current protocol is expected to provide a set of candidate biomarkers reflecting clinically manifest IHD, its severity and therapeutic responses. Furthermore, a long-term follow-up of the study cohorts without IHD (AVR III and CCTA IV) rises to respond to the unmet need of subclinical IHD biomarkers by providing a valuable window of opportunity to assess the time course of IHD development from an epitranscriptomic point of view. In addition, such follow-up of the IHD cohorts (STEMI I and CABG II) can be expected to provide biomarkers with prognostic properties. Moreover, the ability of epitranscriptomics-based blood biomarkers to reflect the responses to both current and novel cardiovascular therapies may extend their clinical applicability even further [114,115,173].

## 5. Conclusions

In conclusion, the IHD-EPITRAN study with its IHD vs. non-IHD cohort design evaluates transcriptome-wide alterations in patients’ blood RNA m^6^A and A-to-I profiles for discovery of IHD biomarkers and druggable targets. Consequently, this main goal of the IHD-EPITRAN study will be achieved via recruitment of patients from four distinct clinical cohorts, two with IHD and two without IHD, all with carefully assessed coronary status. The collected sample RNAs will be analyzed with the state-of-the-art contemporary, both quantitative and qualitative methods, for epitranscriptomic modifications in a seamless collaborative manner with experts from many disciplines, from specialist physicians to methodological experts.

## 6. Contact Us

To further increase its scientific and clinical impact, the IHD-EPITRAN is seeking new clinical and research centers to join the IHD-EPITRAN Consortium. To contact us, the IHD-EPITRAN Consortium has set up a web page (www.ihd-epitran.com) that describes the Consortium members and provides contact information in more detail. The study can be followed on social media (Twitter: @IHD_EPITRAN) and direct inquiries regarding participation should be addressed via email (ihd.epitran@gmail.com).

## Figures and Tables

**Figure 1 ijms-22-06630-f001:**
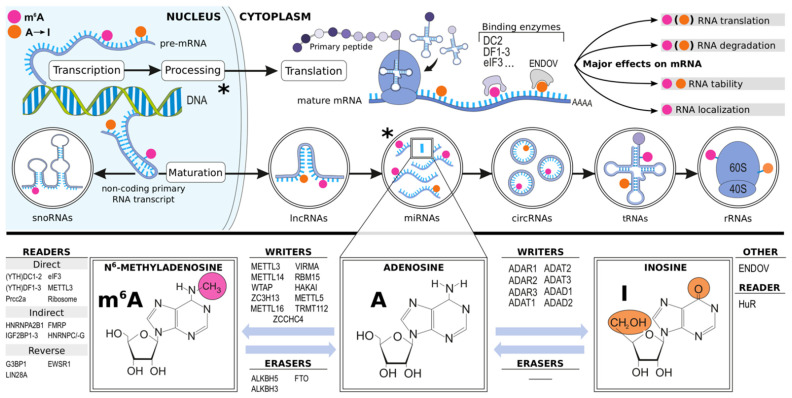
Overview of the RNA m^6^A modification and A-to-I RNA editing and their known writers, readers, and erasers. m^6^A modification and A-to-I editing occur in most RNA species. *ADAD1-2*, adenosine deaminase domain-containing protein 1-2; *ADAR1-3*, double-stranded RNA-specific adenosine deaminase 1-3; *ADAT1-3*, adenosine deaminases acting on tRNAs; *ALKBH5*, alkB homolog 5 RNA demethylase; *A-to-I,* adenosine-to-inosine RNA editing; *circRNA*, circular RNA; *ENDOV*, human endonuclease V; *eIF3*, eukaryotic initiation factor 3; *EWSR1*, Ewing sarcoma breakpoint region 1 protein; *FMRP*, fragile X retardation protein; *FTO*, fat mass and obesity associated protein; *G3BP1*, Ras GTPase-activating protein-binding protein 1; *HAKAI*, E3 ubiquitin-protein ligase Hakai; *HNRNP-A2B1,-C,-G*, heterogeneous nuclear ribonucleoprotein A2/B1 and C1/C2 and G; *HuR*, human antigen R; *IGF2BP1-3*; The insulin-like growth factor-2 mRNA-binding proteins 1, 2, and 3; *LIN28A*, Lin-28 homolog A; *lncRNA*, long non-coding RNA; *METTL3,-14,-16*, N6- adenosine-methyltransferase catalytic subunit/non-catalytic subunit/METTL16; *METTL5*, methyltransferase Like 5; *mRNA*, messenger RNA; *miRNA*, microRNA; *Prcc2a*, proline rich coiled-coil 2 A; *RBM15*, RNA binding motif protein 15; *rRNA*, ribosomal RNA; *snoRNA*, small nucleolar RNA; *TRMT112*, TRNA methyltransferase subunit 11-2; *tRNA*, transfer RNA; *VIRMA*, vir like m^6^A methyltransferase associated; *WTAP*, Wilm’s tumor associated protein; *(YTH)DC1-2,* YTH domain-containing protein 1 ja 2; *(YTH)DF1-3*, YTH N6-methyladenosine RNA binding protein 1-3; *ZCCHC4*, zinc finger CCHC-type containing 4; *ZC3H13*, zinc finger CCCH domain-containing protein 13; *, miRNAs can also derive from pre-mRNA introns.

**Figure 2 ijms-22-06630-f002:**
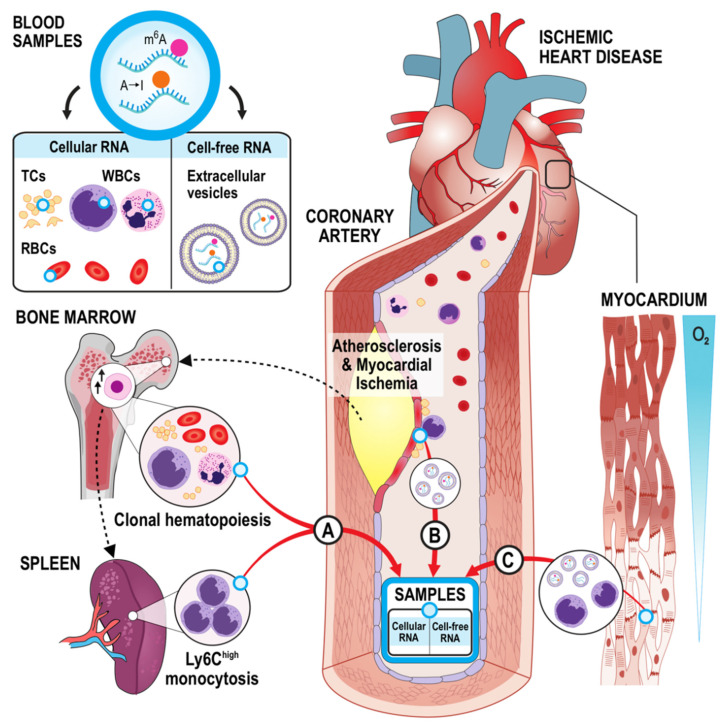
IHD-EPITRAN hypotheses (**A**). Coronary plaques signal bone marrow residing HSCs to increase proliferation promoting the efflorescence of CH and extramedullary hematopoiesis (Ly-6C^high^ monocytosis), which both seed epitranscriptomically distinct cells to the circulation. (**B**). Leukocytes and platelets patrolling in the proximity and inside the atherosclerotic plaques, ischemic myocardium, and stressed endothelium oscillate back and seed EVs to the circulation with detectable alterations in their m^6^A and A-to-I RNA signatures. (**C**). The ischemic myocardium prime patrolling leukocytes and secrete paracrine EVs encasing m^6^A and A-to-I modified RNA molecules, entering also to the circulation. *A-to-I*, adenosine-to-inosine; *CH*, clonal hematopoiesis; *EV*, extracellular vesicle; *HSC*, hematopoietic stem cell; *Ly6C*, lymphocyte antigen 6; *m^6^A*, N^6^-methyladenosine; *RBCs*, red blood cells; *TCs*, thrombocytes; *WBCs*, white blood cells.

**Figure 3 ijms-22-06630-f003:**
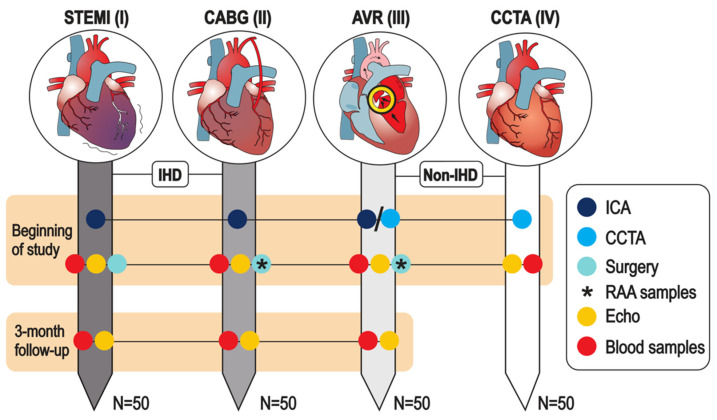
Outline and sample collection in the IHD-EPITRAN study. The gray scale provides an arbitrary scale for disease severity across cohorts. *STEMI*, ST-elevation myocardial infarction; *CABG*, coronary artery bypass grafting; *AVR*, aortic valve replacement; *CCTA*, coronary computed tomography angiogram; *ICA*, invasive coronary angiography; *RAA*, right atrial appendage; *IHD*, ischemic heart disease.

**Figure 4 ijms-22-06630-f004:**
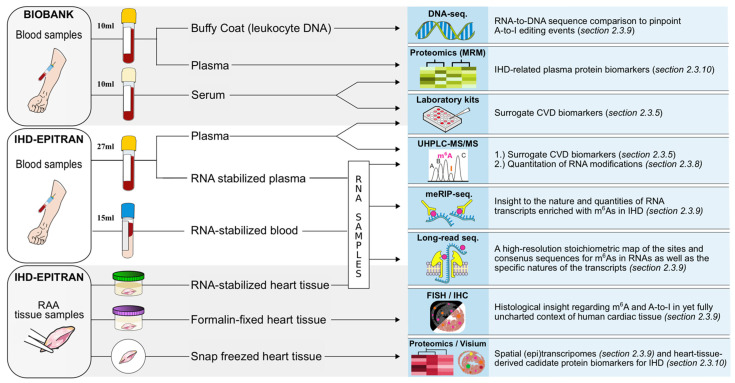
Study samples with respective principal analysis methods of the IHD-EPITRAN study. *CVD*, cardiovascular disease; *FISH*, fluorescence in situ hybridization; *IHC*, immunohistochemistry; *IHD*, ischemic heart disease; *meRIP seq*, methylated RNA immunoprecipitation sequencing; *MRM*, multiple reaction monitoring; *RAA*, right atrial appendage; *UHPLC-MS/MS*; ultra-high-performance triple quadrupole liquid chromatography tandem mass spectrometry.

**Figure 5 ijms-22-06630-f005:**
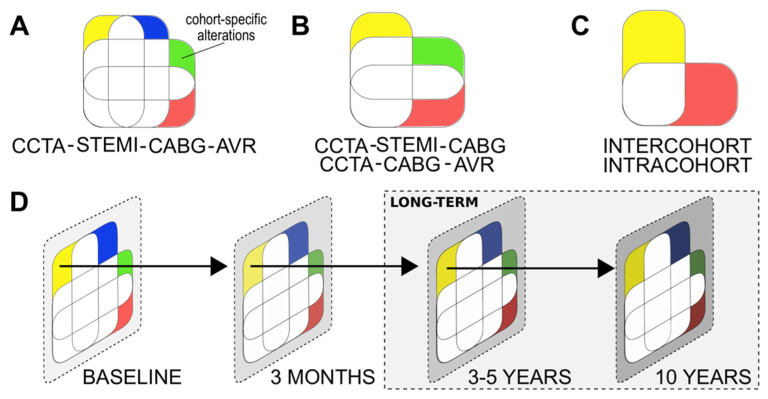
Principal study cohort comparisons in the IHD-EPITRAN study. Referred outcomes are listed in Table 3 (**A**). Venn diagram-based illustration of the preferred four-partite approach, due to its highest degree of adjustments, to acquire inter-cohort comparison-based outcomes. (**B**). Three-partite comparison scheme to enable comparisons even in the possible case of delay in one cohort recruitment, which is also the case with (**C**) depicting pairwise comparisons with lest adjustments for inter-cohort outcomes. Intracohort outcomes are achieved via pairwise prospective comparisons. (**D**). Timeline for the prospective outcome comparisons. Long-term follow-up of the study cohorts I-IV could provide dimensions of detecting incident and IHD exacerbations (Section 2.6 and Discussion). Abbreviations: *AVR*, aortic valve replacement cohort III; *AVS*, aortic valve stenosis; *CABG*, coronary artery bypass grafting cohort II; *CCTA*, coronary computed tomography angiogram cohort I; *CVD*, cardiovascular disease; *IHD*, ischemic heart disease, *MACCE*, major adverse cardiovascular and cerebrovascular event; *PCI*, percutaneous coronary intervention; STEMI, ST-elevation myocardial infarction cohort I.

**Table 1 ijms-22-06630-t001:** Main goals of the IHD-EPITRAN study with brief respective descriptions. *IHD*, Ischemic heart disease.

Goal	Description
To identify IHD-specific candidate biomarkers and lay the foundation for both clinical and diagnostic efficacy studies in the future	*Corss-sectional cohort study design* Comparsion of blood RNA modifications of IHD cohorts to non-IHD cohorts *Porspective cohort study design* Comparsion of circulating RNA modifications of IHD-positive cohorts to thoes in IHD-negative cohort
To establish protocals and modify detection methods and workflows for RNA modications	Optimization of RNA isolation, type-fragmentation for modification-targeted RNA sequencing using both second and third generation methodologies
To increase the pathophysiologic knowledge of IHD	The cross-sectional and prospective comparsions of blood and right atrial appendage epitramscriptomes is expectable to provide novel insight to the IHD pathophysiology
To open venues for therapeutic development previously been disregarded due to the lack of research and methodological limitations	Identification of novel epotanscriptomic candidate blood biomarkers for IHD can provide potenial targets also for future drug development

**Table 2 ijms-22-06630-t002:** General and cohort-specific exclusion criteria for the IHD-EPITRAN study. *CCS*, Canadian cardiovascular society (for angina pectoris grading); *GFReEPI*, Glomerular filtration rate estimated with *CKD-EPI* (Chronic Kidney Disease Epidemiology Collaboration) equation; *IHD*, ischemic heart disease; *LVEF*, left ventricular ejection fraction; *LVH*, left ventricular hypertrophy; *NYHA*, New York Heart Association (for heart failure grading); *PCI*, percutaneous coronary intervention; *RR*, Scipione Riva-Rocci (eponym for sphygmomanometric blood pressure gauge).

General	Justification
Condition that limits life expectancy	May modify blood epitranscriptomes hampering reliable biomarker identification
Active inflammatory state (i.e., gout, systemic lupus erythematosus)	Cimplement, cytokines and leukocyte activation may infuence blood epitranscriptomes
Primary desease of blood or bone marrow	Expectable to alter epotranscriptomic regulation precluding reliable biomarker discovery
Major congenital heart disease of atrial fibrillation	To exclude the effects of remodelling in atrial appendges
Renal insufficiency (*GTReEPI < 45 mL/min*)	To exclude the effects of altered blood solute dynamics for reliable biomarker discovery
Uncontrolled hypertension (*RR > 180/100 mmHg*) or diabetes (*HbA1c > 60 mmol/L*), insulin us diabetes	Hight blood pressure and significant hyperglycemia damage endothelium and blood cells
Prior open-heart surgery (i.e., coronary artery bypass surgery)	To exclude very high-morbidity IHD and support the goal to identify biomarkers for early-to-moderate IHD
Other manifestations of atherosclerosis: a. Arteriosclerosis oblitierans/claudication b. Earlier stroke, cerebral hemorrhage or transient ischemic attach (TIA) c. Vascular or mixed type dementia d. Clinically releveant carotid artery stenosis e. Mesenteric ischemia	To exclude effects of other manifestations of atherosclerosis in blood epitranscriptomes as extensively as possible; Except for surveying vascular claudication, prospective investigations to exclude these manifestations will not be performed; Ankle-brachial index is recorded for any asymptomatic peripheral artery disease in cohort II
Transthoracic echocardiography: a. Cardiomyopathy (Hypertrophic/Dilated) b. Left ventricular hypertrophy (LVH) c. Clear heart failure (*i.e., LVEF < 25%)* d. Indication of atrial remodeling e. Functionally significant valve defects	To exclude remodeling effects due to intrinsic myocardial pathology, significant heart failure or valve defects; LVH is considered as an exception for the cohort III (part of pathophysiology)
**Study cohort I, patients with myocardial infarction revascularized with urgent PCI**	
Stent thrombosis, vasospastic coronary occlusion	This is IHD-focused study, myocardial infartions of other than atherothrombotic etiology are excluded
MI complications (*e.g., chordal rupture, aorthic disserction, acute heart failure, cardiogenic shock*)	To focus biomarker discovery to the IHD-induced infarction-specific epitranscriptomic alterations
Global ischemia on electrocardiogram	High rish of insufficient PCI and ischemia relievement
**Study cohort II, patients with stable IHD undergoing coronary artery bypass surgery**	
Duration of stable angina pectoris or exertional dyspneas < 1-month, crescendo angina	To exclude acute and subacute IHD related alterations in blood epitranscriptomes
Complex surgeries (e.g., valve/aneurysm repair)	To exclude other major cardiac remodeling effects
**Study cohort III, patients with stable aortic valve stenosis undergoing valve replacement surgery**	
Clinically mild-to-moderate stenosis with mild symptoms (*NYHA/CCA 0-I*)	Indicate less pronounced pathophysiology that might be reflected with blunted changes in the alterations of the blood epitranscriptomes regarding AVS
Documented IHD or complex operations	As a control cohort with non-IHD cardiac pathology, any indication of IHD will lead to exclusion
Transcatheter asortic valve implantations	To enable right atrial appendage sample collection
**Study cohort IV, patients screened negative for IHD with coronary computed tomography**	
Any prior cardiovascular disease or medication currently or in history	As critical non-IHD controls, the aim is to also recruit patients that represent overall “cardiovascular health”

**Table 3 ijms-22-06630-t003:** Planned outcomes of the IHD-EPITRAN study. Outcome 1 and outcomes 2–5 encase inter- and intra-cohort comparisons, respectively. Morbidity parameters: Section 2.3.2. *A-to-I*, adenosine-to-inosine; *AVR*, aortic valve replacement (III); *AVS*, aortic valve stenosis; *CABG*, coronary artery bypass grafting (II); *CCTA*, coronary computed tomography angiogram (IV); *m^6^A*, N^6^-methyladenosine; *IR*, ischemia-reperfusion; *PCI*, percutaneous coronary intervention; *STEMI*, ST-elevation MI (I).

Primary	Secondary
1. Study sample m6A and A-to-I profiles from the recruitment stage associating with IHD and AVS pathophysiology 1.1. Acute IR controlled for stable ishcemia, pressure overload, and homeostasis (I vs. II-III-IV) 1.2. Stable ishcemia controlled for acute IR, pressure overload, and homeostasis (II vs. I-III-IV) 1.3. Acute IR controlled for homeostasis (I vs. IV) 1.4. Pressure overload controlled for acute IR, stable ishcemia, and hemeostasis (III vs. I-II-IV)	4. Changes in the study sample m^6^A and A-to-I profiles (Recruitment vs. 3-months) associating to 4.1 Therapy effects on Pathophysiology at 3-month follow-up 4.1.1. STEMI-PCI effects (Resolution and relievement acute IR, remodelingl I vs. I) 4.1.2. IHD-CABG effects (Relievement of stable ischemia, remodeling; II vs III) 4.1.3. AVS-AVR effects (Relievement of pressure overload, remodeling; III vs. III) 4.2 All-cause mortality during 3-month follow-up 4.3 Beneficial/adverse/no-response for therapy as measured via echocardiography at 3 months 4.4 Surrogate (non-)metabolite CVD biomarker levels at recruitment, 3 months, changes between
2. Study sample m^6^A and A-to-I profiles from the recruitment stage associating with 3-moth follow-up clinicial parmeters 2.1. Cardiovascular mortality 2.2 Cardiovascular morbidity 2.3. MACCE 2.4 Cardiovascular medication increases or reductions	5. study sample m^6^A and A-to-I profiles from the recruitment stage as in outcomes 4.2.–4.5.
3. Changes in the sample m^6^A and A-to-I profiles (recruitment vs. 3-months) as in outcome 2	6. Study sample quantitative alterations in other RNA modifications as in outcomes 1–5 (2.3.8.)
	7. Plasma metabolute and non-metabolite CVD biomarkers as in outcomes 1–5 (excl.4.4) (2.3.5.)
	8. Biobank plasma and RAA sample proteomic profiles as in outcomes 1–5 (2.3.10.)
	9. Whole blood sample transcriptome-based leukpcyte profiles as in outcomes 1–5

**Table 4 ijms-22-06630-t004:** Current controller and collaborator centers of the IHD-EPITRAN study with respective main responsibilities. *A-to-I*, Adenosine-to-inosine RNA editing; *AVR*, Aortic valve replacement surgery cohort III; *AVS*, aortic valve stenosis; *CABG*, coronary artery bypass surgery cohort II; *CCTA*, coronary computed tomography angiogram cohort IV; *ICA*, invasive coronary angiography; *meRIP-seq*, methylated RNA immunoprecipitation sequencing; *m^6^A*, N^6^-methyladenosine; *PCI*, percutaneous coronary intervention; *TTE*, transthoracic echocardiography.

**Controllers**	**Main responsibility area/task**
Heart and Lung Center & Cardiac Unit, Helsinki University Hospital (HUS), Finland	Patient recruitment; clinicial evaluations; operations (PCA, CABG, AVR); imagings (ICA, CCTA, TTE); control visits; study sample collection; registry data storage and governace; sample storage
Department of Pharmacology, University of Helsinki (UH), Finland	Coordination of (1) collaboration, (2)funding acquisition and (3) competitve tendering; RNA exctraction and initial sample quality measurements; academic analyses; scientific publishing
**Collaborator centers**	
Heart Hospital, Tampere Unviersity Hospital (TAYS), Finland	Patients recruitment (cohort III); clinical evaluations; operations (AVR); imagings (ICA, TTE); control visits; sample collection; clinical data and sample storage
Helsinki and Tampere Biobanks, Finland	Additional biobank sample collection and storage; DNA extraction; protocal designing
Meilahti Clinical Proteomics Core Faculity, UH, Finland	Targeted proteomics from plasma and snap-frozen RAA tissue samples; protocal designing
Chemistry Unit, Ginnish Food Authority, Finland	Specific measurements and analyses of modified RNA necleotides from via UHPLC-MS.MS methodolgy
Folkhälsan Research Center, Finland	Leading bioinformatics of second and third generation RNA sequencing targeting modifications (m^6^A) as well as DNA-to-RNA sequence matching (A-to-I)
Middle East Technical Univeristy, Ankara, Turkey	Ollaboration in the bioinformatics with Folkhälsan, specific share of responsiblities is decided later
Koç University, Istanbul, Turkey	See above
University of Tartu, Estonia	Leading the collaboration for the development of potential binding molecules as novel drugs for IHD

**Table 5 ijms-22-06630-t005:** Consideration of the strengths and limitations of the IHD-EPITRAN study. *ABI*, ankle-brachial index, *ATR*, anatomical therapeutic chemical classification system; *AVR*, aortic valve replacement cohort III; *AVS*, aortic valve stenosis; *CABG*, coronary artery bypass grafting cohort II; *CCTA*, coronary computed tomography angiogram cohort IV; *CVD*, cardiovascular disease; *DDT*, defined daily dose, *EV*, extracellular vesicle; *ICA*, invasive coronary angiography; *IHD*, ischemic heart disease; *PAD*, peripheral artery disease; *TTE*, transthoracic echocardiography.

**Strengths**
1. Interdisciplinary collaboration enables simultaneous recruitment, sample handling and measurements.
2. A relevant IHD vs. non-IHD comparison is incorporated with a follow-up dimension as well.
3. Coronary artery status is visualized from all participants recruited.
4. Cardiac function is assessed by TTE from all participants recruited.
5. Exclusion of patients with prior clinically relevant manifestations of atherosclerosis than IHD.
6. Participants are surveyed for vascular claudication. ABI for the cohort II to record any asymptomatic PAD.
7. Cohort morbidity an dother background characteristics are meticulously registered, assessed and reported.
8. Study sample collection is designed to minimize the timespan for RNA vulnerable for degradation.
9. Sample use is optimized for as comprehensive measurements as possible. RNA is fractionated for analyses.
**Limitations and management consideration**
1. Coronary evaluation is non-uniform across cohorts. *SYNTAX with ICA for complexity assesment of IHD. The CCTA cohort IV is a primary non-IHD control.*
2. Complete rule out of the effects of systemic atherosclerosis cannot be achieved *Most earlier manifestations of systemic atherosclerosis lead to exclusion, vascular claudication is surveyed (exclusion criterion) and ABI is measure from CABG cohort to record any asymptomatic PAD.*
3. Limited possiblity to adjust medication effects (cohort IV has neither any CVD pathology nor medication). *ATC and DDT alterations are recorded and reported. Immunomodulatry, edication leads to exclusion.*
4. Limited ability to pinpoint precise origins of the forthecoming epitranscriptomic alterations. *Buffy coat leukpcytes can be used for validation, plasma cfRNA is principally derived from EVs.*
5. Incapability to identify biomarkers directly for subclinical IHD due to required cohort morbidity. *(1) Current study assesses the harnessing potential of epitranscriptomics for a source of IHD biomarkers* *(2) Long-term follow-up of cohort IV can address the task.*

## Data Availability

Not applicable (however, see Section 2.4).

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
