# Peer review of "Epitranscriptomics of Ischemic Heart Disease—The IHD-EPITRAN Study Design and Objectives"

_ijms, 2021, doi:10.3390/ijms22126630_

Round 1
Reviewer 1 Report
The Study design article entitled: "Epitranscriptomics of ischemic heart disease—the IHD-EPI-TRAN study design and objectives" is a schematic and well written article on the role of the epitranscriptomic in IHD. The IHD-EPITRAN study can be expected to enable identification of epitranscriptomic IHD biomarker candidates and potential drug targets.
The topic is interesting, comprehensive and meaningful. The figures and Tables are clearly presented, and references are updated and adequate.
The topic is perfectly in line with the "IJMS" journal and therefore I recommend the publication of the manuscript.
Reviewer 2 Report
In this manuscript, Sikorki et al discuss the novel hypothesis that epitranscript marks in blood-derived RNAs, in particular m6A and A-to-I modifications, could mirror pathological features of IHD. This concept reflects an innovative direction that is of great interest in the field. In addition, the results of this study, by comparing data originating from four cohorts of patients (two with IHD and two without IHD), will be very informative to find potential IHD-specific candidate modified RNAs that can be used as new biomarkers.
The paper is overall well written and organized and the authors extensively describe the experimental plans to prove their hypothesis. In addition, the figures are well planned and very useful to visualize the key concepts.
Overall, I consider this manuscript suitable for publication.
See some minor suggestions below:
In section 1.2 (Line 92) The existence of other epitranscriptomic marks should at least be mentioned (such as m1A, m7G, m5C etc ). Even though it is not the focus of this paper, and overall these marks are less abundant and characterized, it is important to acknowledge the existence of other epitranscript modifications in this introductory paragraph. This is especially important since some of these marks are mentioned as potential future directions in lines 554-555.
The sentence in Line 93 is not precise. Please modify to : “On the molecular level, the bases of a newly transcribed RNA strand undergo extensive modifications both in the nucleus and cytoplasm”. It does not make sense to state that “on the cellular level” RNA bases are modified!
Line 135: Specify better what you mean by “localization in RNA”. This sentence is ambiguous and could have different meanings.
Legend of Figure 3. When explaining abbreviations please follow the same order as in the figure to make it easier for the reader to find the terms (first STEMI, CABG, AVR, CCTA then ICA, CCTA, RRA…)
Line 417 states “The detailed protocol for sequencing will be defined during the course of the study”. However, since this is the central point of the project the authors should carefully evaluate all available options well in advance and describe the specific technology that will be used in more details (although it is implied that this might be subject to change as there are continuously new technical developments in the field)
